# Genome-wide association study to identify the genomic loci associated with wheat heading date variation under autumn-sowing conditions

Yurim Kim[1], Myoung-Goo Choi[1], Myoung Hui Lee[1], Chuloh Cho[1], Jun Yong Choi[2], Suk-Jin Kim[1], Chon-Sik Kang[1], Chul Soo Park[3], Ki-Chang Jang[1], Youngjun Mo[3]*, Changhyun Choi[1]*

**1** National Institute of Crop Science, Wanju, Republic of Korea, **2** Department of Central Area Crop Science, National Institute of Crop Science, Suwon, Republic of Korea, **3** Department of Crop Science and Biotechnology, Jeonbuk National University, Jeonju, Republic of Korea

* chchhy@korea.kr (CC); yjmo@jbnu.ac.kr (YM)

## Abstract

The present study aimed to identify genetic loci associated with days to heading (DTH) in wheat under autumn-sowing conditions in Korea, where early heading is critical owing to the overlap between the wheat harvest and the rainy season. We evaluated 530 wheat core collections over five years, focusing on known heading date genes *VRN-1* and *PPD-1*, and conducted a genome-wide association study (GWAS) to identify new genetic loci related to DTH. The results revealed that Korean accessions exhibited the earliest DTH, with modern Korean varieties heading even earlier, reflecting a strong breeding focus on early heading. Among the existing heading date genes, *VRN-1* and *PPD-D1* were significantly associated with DTH in the wheat core collection. However, all Korean varieties carried the same alleles for each of *VRN-A1*, *PPD-A1*, and *PPD-D1*, resulting in low genetic diversity, which rendered the existing heading date genes insufficient to fully account for the variation in DTH within the Korean varieties. GWAS identified nine single nucleotide polymorphisms (SNPs) associated with DTH in Group A (entire collection filtered, n=518) and six in Group B (accessions with genotypes identical to Korean varieties filtered, n=231). Four key SNPs (AX-95222044 and AX-94685526 in Group A, and AX-94550996 and AX-94970315 in Group B) were selected based on their effect sizes on DTH. In both groups, accessions with alleles for early heading at both of the selected SNPs exhibited the earliest DTH, advancing by 7.7 to 8.9 days. These findings suggest that the selected SNPs, particularly those reflecting the genotypes of Korean varieties, effectively explain the variations in DTH among Korean varieties and could enhance wheat breeding efficiency in Korea. Further research is needed to validate the four selected SNPs and identify the underlying genes, which could serve as valuable markers for developing early-heading wheat varieties suited to Korean autumn-sowing conditions.

**Data availability statement:** All relevant data for this study are publicly available from the Zenodo repository (https://doi.org/10.5281/zenodo.15075033).

**Funding:** This work was supported by a grant from the "Development of winter wheat sowing technology for extension double cropping system in southern area" Program (No. PJ016821012024), Rural Development Administration, Republic of Korea to YK. The funders had no role in study design, data collection and analysis, decision to publish, or preparation of the manuscript.

**Competing interests:** The authors have declared that no competing interests exist.

## Introduction

The heading date is a critical factor in the formation and development of wheat seeds, significantly affecting grain yield and quality. Among numerous genetic factors controlling flowering, genes involved in vernalization (*VRN*), photoperiod (*PPD*), and earliness per se (*Eps*) have been extensively studied in wheat [1,2]. *VRN*, *PPD*, and *Eps* contribute approximately 70–75%, 20–25%, and 5%, respectively, to wheat heading date variations [3,4].

To date, four *VRN* genes (*VRN-1*, *VRN-2*, *VRN-3*, and *VRN-4*) and two *PPD* genes (*PPD-1* and *PPD-2*) have been identified [5–9]. *VRN-A1, VRN-B1*, and *VRN-D1* are located on chromosomes 5A, 5B, and 5D, respectively. *VRN-A2* was mapped to chromosome 5A, while *VRN-A3, VRN-B3*, and *VRN-D3* are located on chromosomes 7A, 7B, and 7D, respectively. Among the *VRN-4* genes, *VRN-D4* is located on chromosome 5D [10–16]. *VRN-1* shows greater variability in response to vernalization than other genes, and *VRN-A1* has a higher responsiveness than *VRN-B1* or *VRN-D1* [1,17]. Spring wheat and facultative wheat, which is adaptable to both spring and winter sowing conditions, do not require vernalization, whereas winter wheat does. Spring wheat carries one or more dominant alleles in *VRN-1*, *VRN-3*, and *VRN-4*, whereas winter wheat carries a dominant allele in *VRN-2* and recessive alleles (*vrn-1*, *vrn-3*, and *vrn-4*) at other loci [1,7,9,18]. *PPD-A1*, *PPD-B1*, and *PPD-D1* are located on chromosomes 2A, 2B, and 2D, respectively. Among *PPD-2* genes, *PPD-B2* is located on chromosome 7BS. Photoperiod response is the strongest for *PPD-D1*, followed by *PPD-B1* and *PPD-A1*. The photoperiod-insensitive type, with dominant alleles (a-type), heads as soon as the temperature rises in spring, whereas the photoperiod-sensitive type, with recessive alleles (b-type), heads only when the day length is sufficiently long [8,9,19–24]. The heading date of spring wheat is significantly influenced by the allelic form of *VRN-1*, whereas that of winter wheat is greatly influenced by the allelic forms of *PPD-D1* and *PPD-B1* [1,25–27]. *Eps* affects heading time in the absence of the effects of vernalization and photoperiod genes. Recently, studies on *EARLY FLOWERING 3 (ELF3)*, a component of the evening complex (EC) in the circadian clock, have shown that *ELF3* loss-of-function results in early heading under both long-day and short-day conditions [28–32].

In addition to the *VRN* and *PPD* genes, active research has been conducted to explore quantitative trait loci (QTLs) for wheat heading date [9,33–35]. Most QTL studies using bi-parental mapping populations, such as $F_2$ or recombinant inbred lines, are dependent on relatively narrow genetic diversity segregating from the two parental lines. In contrast, genome-wide association studies (GWAS) with diversity panels constitute a useful method for constructing a high-resolution genetic map in a relatively short time using resources with diverse genetic compositions [36–38]. To date, many single nucleotide polymorphism (SNP) markers associated with the heading date variation have been reported on most of the wheat chromosomes [13,14,34,35,39–46].

In Korea, which experiences a summer monsoon climate, the wheat harvesting season may overlap with the rainy season [11,47]. Frequent rainfall before harvest can lead to preharvest sprouting and lodging, causing delays in wheat

harvesting operations and potential quality degradation [48]. Therefore, to ensure stable yield and quality of wheat, advancing the heading date would be necessary to accelerate maturation without shortening the duration of the maturation phase. To enhance the selection efficiency of wheat breeding programs for the development of early maturing varieties, analysis of the genetic factors controlling heading date variation under the conditions of autumn sowing, prevalent in these regions, is necessary. In this study, we evaluated variations in the heading date of 530 wheat accessions of various origins under autumn-sowing conditions in Korea for five years. By determining the allele types of *VRN-1* and *PPD-1* and conducting GWAS analysis, we identified new genetic loci associated with the heading date, providing useful information for breeding early-heading wheat varieties suitable for the Korean climate under autumn-sowing conditions.

## Materials and methods

### Plant materials

This study used 530 wheat genetic resources and an additional 40 Korean varieties as experimental materials. The 530 accessions belong to a wheat core collection previously established for genomic analysis [49]. Among these accessions, 120 were collected from Korea, and 362 were collected from 47 countries including Mexico (101), the USA (42), China (31), Ethiopia (22), Afghanistan (13), Russia (12), Canada (10), and Turkey (10), while the origin of 48 accessions is unknown (S1 Table).

### Heading date investigation

The plant materials were sown on October 25th each year from 2018 to 2022 at the National Institute of Crop Science (Jeollabuk-do, Republic of Korea) and cultivated according to the Rural Development Administration (RDA) standard cultivation method [50]. They were grown according to the ear-to-row method, where seeds from a single ear are sown in a single row, with each accession planted in three 1.5 m rows spaced by 0.4 m. The fertilization was based on N:$P_2O_5$:$K_2O$ = 9.1:7.4:3.9 kg/10a. $P_2O_5$ and $K_2O$ were applied at the time of sowing, while N was divided into two applications, with 40% at sowing and 60% at the regeneration stage, the period of regrowth after winter. The heading date was determined when approximately 40% of the total number of heads had emerged, and the days to heading (DTH) from sowing were counted.

The average temperature during the test years (2018–2022), from sowing (October 25th) to the elongation stage (late April) before heading, was 6.3°C, which was 0.6°C higher than the long-term average (1991–2020) (S1 Fig). The average accumulated precipitation during the same period over the test years was 270 mm, which was 21 mm less than the long-term average.

### Genomic DNA extraction and PCR analysis

Leaves of plants grown in a greenhouse for three weeks were rapidly frozen in liquid nitrogen, ground into a fine powder using a mortar and pestle, and then stored at –70°C. Genomic DNA was extracted from the leaf powder using a plant genomic DNA extraction reagent (Solgent Co., Korea). Extracted genomic DNA was quantified using a Nanodrop 1000 spectrophotometer (Thermo Fisher Scientific, Waltham, MA, USA) and stored at –20°C for use. Genotyping analysis of *VRN-A1*, *VRN-B1*, and *VRN-D1* was performed according to the methods of Whittal et al. [51] and Fu et al. [13], and *PPD-A1*, *PPD-B1*, and *PPD-D1* were analyzed using the methods outlined by Plotnikov et al. [52] and Beales et al. [53] (S2 Table). The stability of DTH across years was assessed for each genotype using Shukla's Stability Variance (SSV), as described by Shukla [54]. A linear mixed model (LMM) was applied to partition the phenotypic variance into components attributable to genotype, environment, and genotype-environment interaction (G×E). In the model, genotypes were treated as fixed effects, while years were treated as random effects. SSV values were calculated based on the variance of G×E interactions, with lower SSV values indicating greater stability.

## Genome-wide association studies (GWAS)

For GWAS, genotype data of the 530 wheat accessions from the 35K SNP iSelect chip array (Thermo Fisher Scientific) were utilized [55]. From the initial 35,143 SNPs, 3,218 that are not located on the chromosomes of the IWGSC v1.0 reference (CerealsDB) were excluded. To utilize only high-quality SNPs for GWAS, the filtering process was conducted following the method described by Marees et al. [56]. To minimize redundancy, SNPs were excluded when the calculated $R^2$ value between markers within a 50 kb region exceeded 0.2 through LD pruning [57]. Moreover, accessions with a heterozygosity ratio significantly higher than the average, exceeding +3 standard deviations (0.4327161), were excluded from the analysis. Ultimately, 10,152 SNPs and 518 accessions were used for principal component analysis (PCA) and GWAS.

The initial GWAS was conducted using 518 accessions (Group A). Subsequently, a second GWAS was performed using only 231 accessions (Group B) possessing the *vrn-A1*, *Ppd-A1b*, and *Ppd-D1a* alleles (the same allelic composition as Korean wheat varieties). GWAS was conducted using Genome Association and Prediction Integrated Tool (GAPIT) version 3 [58], applying six methods: Mixed Linear Model (MLM), Compression MLM (CMLM), Bayesian-information and Linkage-disequilibrium Iteratively Nested Keyway (BLINK), Multi-Locus Mixed-Model (MLMM), Fixed and random model Circulating Probability Uniform (FarmCPU), and Settlement of MLM Under Progressively Exclusive Relationship (SUPER).

After the GWAS, SNPs with $-\log_{10}(P) > 5$, MAF (minor allele frequency) > 5%, and an effect size of at least 3.9 days on DTH were selected. Information on candidate genes near the selected SNPs was obtained from Ensembl Plants, and gene expression was verified using the Wheat Expression Browser (RefSeq1.1). Gene functions were also estimated with reference genomes to the Rice Annotation Project and TAIR (The Arabidopsis Information Resource).

## Statistical analysis

All statistical analyses were performed using R software version 4.3.1 (R Core Team, 2023; Vienna, Austria). For the analysis of heading date phenotypes, Duncan's multiple range test was conducted using the "agricolae" package. The "psych" package was employed to visualize the correlation between data across different years. Broad-sense heritability was calculated using the "variability" package. Analysis of variance (ANOVA) and *t*-tests were performed using the "stats" package.

## Results

### Heading date variation of the core collection and Korean wheat varieties

In the wheat core collection comprising 530 accessions from 47 countries, the broad-sense heritability of DTH was high at 0.89 (S3 Table). The average DTH during the test years (2018–2022) was 188.6 days, and there was a high correlation (r ≥ 0.95***) between the average DTH and the DTH in each test year (S2 Fig). Among the top five countries of origin comprising the wheat core collection, accessions from Korea exhibited the earliest DTH at 183.3 days, followed by those from Ethiopia (186.3 d), China (186.8 d), Mexico (186.7 d) and the USA (192.8 d) (Fig 1). In addition, Korean varieties (n = 40; modern cultivars released in 1976–2016) exhibited an even earlier DTH of 178.5 days, on average, which was 4.8 days earlier than the Korean landrace accessions (n = 120) in the wheat core collection. This distribution of DTH indicated that accelerating heading has been a major goal in Korean wheat breeding programs.

### Allelic variations of *VRN-1* and *PPD-1* in the wheat core collection and Korean varieties

To characterize the allele types of previously known major heading date genes, genotyping experiments were conducted using molecular markers for *VRN-1* and *PPD-1* (Table 1). In the wheat core collection (n = 530), 410 (77.4%) accessions carried the winter-type *vrn-A1*, while 93 (17.5%) accessions carried the spring-type *Vrn-A1*. The DTH for accessions carrying the *Vrn-A1* allele averaged 189.5 days, which was significantly longer (*p < 0.05*) than the average of 188.4 days for accessions carrying *vrn-A1*. For *VRN-B1*, 333 (62.8%) accessions carried the winter-type *vrn-B1* and

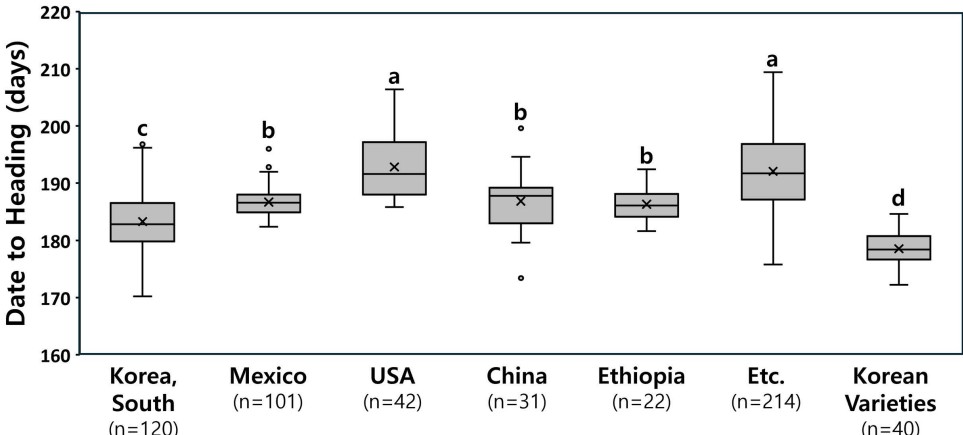

**Fig 1. Distribution of days to heading in the wheat core collection (n = 530) by geographic origin and Korean wheat varieties (n = 40).** Different letters above the boxplots indicate significant differences at *P* < 0.05 level according to Duncan's Multiple Range Test.

**Table 1. Effects of *VRN-1* and *PPD-1* Alleles on days to heading (DTH) in the wheat core collection.**

| Gene | Allele | Number of accessions | Average of DTH | Shukla's Stability Variance |
|---|---|---|---|---|
| *VRN-A1* | *Vrn-A1* | 93 (17.5%) | 189.5* | 1.27 |
| | *vrn-A1* | 410 (77.4%) | 188.4 | 0.83 |
| | unknown | 27 (5.1%) | 187.3 | 1.62 |
| *VRN-B1* | *Vrn-B1* | 173 (32.6%) | 189.4* | 0.59 |
| | *vrn-B1* | 333 (62.8%) | 188.3 | 1.03 |
| | unknown | 24 (4.5%) | 186.7 | 2.31 |
| *VRN-D1* | *Vrn-D1* | 84 (15.8%) | 187.0 | 1.19 |
| | *vrn-D1* | 330 (62.3%) | 189.8*** | 0.8 |
| | unknown | 116 (21.9%) | 186.1 | 1.3 |
| *PPD-A1* | *Ppd-A1a* | 4 (0.8%) | 189.4 | 4.03 |
| | *Ppd-A1b* | 521 (98.3%) | 188.6 | 0.87 |
| | unknown | 5 (0.9%) | 189.3 | 1.99 |
| *PPD-B1* | *Ppd-B1b* | 506 (95.5%) | 189.0 | 0.82 |
| | unknown | 24 (4.5%) | 180.0 | 6.34 |
| *PPD-D1* | *Ppd-D1a* | 290 (54.7%) | 185.4 | 1.69 |
| | *Ppd-D1b* | 203 (38.3%) | 192.5*** | 0.66 |
| | unknown | 37 (7.0%) | 191.8 | 0.56 |

ʸ**p* < 0.05, ***p* < 0.01, ****p* < 0.001.

173 (32.6%) carried the spring-type *Vrn-B1*. Accessions carrying *Vrn-B1* exhibited a significantly longer DTH of 189.4 days (*p* < 0.05) than the 188.3 days for accessions with *vrn-B1*. For *VRN-D1*, 330 (62.3%) and 84 (15.8%) accessions carried the winter-type *vrn-D1* and the spring-type *Vrn-D1*, respectively. Accessions carrying *vrn-D1* exhibited a significantly longer DTH of 189.8 days (*p* < 0.001) than the 187.0 days for accessions with *Vrn-D1*. For *PPD-A1*, 521 (98.3%) accessions carried the photoperiod-sensitive type *Ppd-A1b*, while 4 (0.8%) carried the photoperiod-insensitive type *Ppd-A1a*, with no significant difference in DTH. For *PPD-B1*, no accession was found to carry the photoperiod-insensitive type *Ppd-B1a*, whereas 506 accessions (95.5%), for which genotype testing was completed, were confirmed

to carry the photoperiod-sensitive type *Ppd-B1b*. For *PPD-D1*, 290 (54.7%) accessions carried the photoperiod-insensitive type *Ppd-D1a*, and 203 (38.3%) carried the photoperiod-sensitive type *Ppd-D1b*. The DTH for accessions carrying *Ppd-D1a* was 185.4 days, which was significantly earlier ($p < 0.001$) than the 192.5 days for accessions carrying *Ppd-D1b*. Allele types of all six genes (*VRN-A1*, *VRN-B1*, *VRN-D1*, *PPD-A1*, *PPD-B1*, and *PPD-D1*) were determined in 329 out of the 530 accessions, exhibiting 16 different allele combinations (S4 Table). The DTH was significantly earlier for combinations with *Ppd-D1a*, and no clear trend was observed for *VRN-1* allele combinations. Based on these results, the effect of the interaction between *PPD-D1* and *VRN-1* genotypes on DTH was analyzed. In the photoperiod-insensitive *Ppd-D1a* allele, the spring-type alleles of *VRN-A1* and *VRN-B1* exhibited significantly longer DTH than that of the winter-type alleles ($p < 0.01$ and $p < 0.001$, respectively). However, for *VRN-D1*, no significant difference in DTH was observed between the spring-type and winter-type alleles. Similarly, in the photoperiod-sensitive *Ppd-D1b* allele, there was no statistically significant difference in the DTH between the spring-type and winter-type alleles of *VRN-1*. Korean accessions, which exhibited the earliest DTH, had a higher proportion of photoperiod-insensitive *Ppd-D1a* than those of other countries, but a relatively lower proportion of the spring alleles of *VRN-1* (S5 Table). The accessions from the USA, which exhibited the latest DTH, had higher proportions of winter-type *vrn-D1* and photoperiod-sensitive *Ppd-D1b* than those of other countries.

All 40 Korean varieties carried *vrn-A1* for *VRN-A1* (S5 Table). For *VRN-B1*, 32 (80.0%) varieties carried *vrn-B1* and 8 (20.0%) carried *Vrn-B1*, with no significant difference in the DTH between the genotypes. For *VRN-D1*, 23 (57.5%) varieties carried *vrn-D1* and 17 (42.5%) carried *Vrn-D1*, with no significant difference in the DTH between the genotypes. For *PPD-A1*, all varieties carried *Ppd-A1b*, whereas for *PPD-D1*, all varieties carried *Ppd-D1a*. For *PPD-B1*, 31 (77.5%) varieties carried *Ppd-B1b*, and 9 (22.5%) carried *Ppd-B1a*. Contrary to expectations, varieties with photoperiod-sensitive *Ppd-B1b* exhibited significantly earlier DTH (177.9 days) than those with photoperiod-insensitive *Ppd-B1a* (180.7 days). Korean wheat varieties had higher proportions of spring and photoperiod-insensitive alleles for *VRN-B1, VRN-D1, PPD-B1*, and *PPD-D1* than those of the Korean accessions in the core collection (S5 Table).

## GWAS on days to heading

PCA was performed on 518 accessions (12 accessions excluded through the genotype filtering) using 10,152 filtered SNPs. The results showed that the first, second, and third principal components (PC1, PC2, and PC3) explained 6.0%, 3.8%, and 2.7% of the total genetic variance, respectively. PCA revealed that Korean accessions clustered distinctly from other accessions, especially distant from Mexican accessions (S3 Fig).

GWAS analyses for DTH were conducted by classifying the core collection into two groups: Group A (n=518) comprised the entire collection, and Group B (n=231) comprised the accessions carrying the *VRN-1* and *PPD-1* genotypes (*vrn-A1*, *Ppd-A1b*, and *Ppd-D1a*) identical to the Korean varieties. As no SNP exceeding the $-\log_{10}(P)$ threshold was detected with the MLM, CMLM, and MLMM models (Fig. 2A), we report the SNPs for DTH identified from the BLINK, SUPER, and FarmCPU models with the threshold of $-\log_{10}(P) > 5$ (Figs 2, 3, S4, and S6 Table). In Group A, nine SNPs were detected on chromosomes 1D, 2A, 2 B, 3A, 4 B, 6D, and 7 B. In Group B, six SNPs were detected on chromosomes 2A, 2 B, 3D, 4A, and 6D. Among the SNPs identified in Group B, AX-94763018 (2A: 38,913,049) was located on the short arm of chromosome 2A, in close proximity to *PPD-A1* (*TraesCS2A02G081900*; 2A: 36,933,684–36,938,202). Considering that all Group B accessions carried the *Ppd-A1b* allele, this SNP presents the potential for a new gene associated with DTH in this region, distinct from *PPD-A1*.

To consider the practical impact of DTH loci on phenotypes, SNPs with an effect size (the difference in DTH between the accessions carrying contrasting genotypes) of at least 3.9 days were selected (Tables 2 and S6). The effect size of 3.9 days corresponds to Cohen's d of 0.62, which is considered to indicate a greater-than-average effect [59,60]. Based on this criterion, AX-95222044 on chromosome 4B and AX-94685526 on chromosome 2B were selected in Group A. The accessions carrying an allele for early heading at AX-95222044 (G SNP) and AX-94685526 (C SNP) headed 4.8 days and

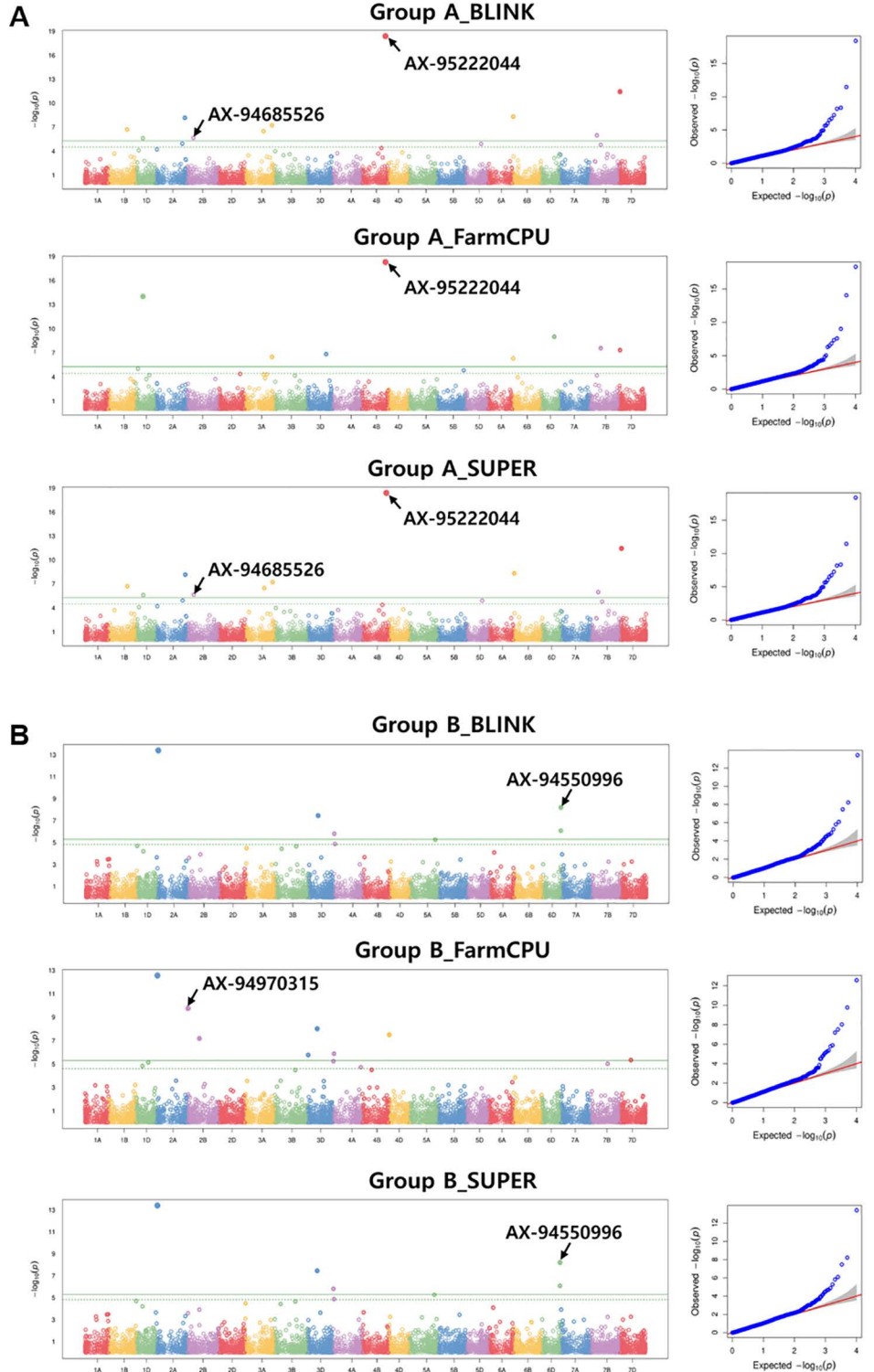

**Fig 2. Manhattan plots and QQ plots for different genome-wide association study (GWAS) models.** The results correspond to the application of the BLINK, FarmCPU, and SUPER models to Group A (A) and Group B (B).

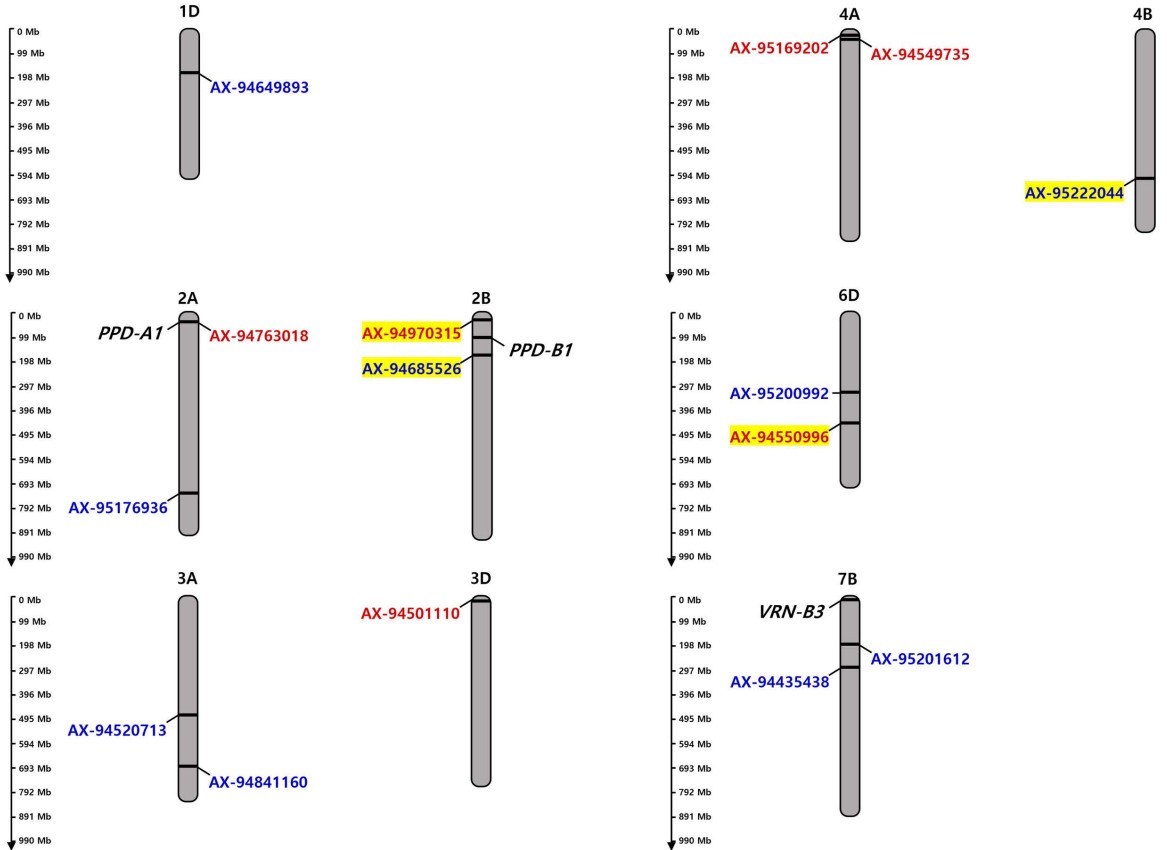

**Fig 3. Chromosomal distribution of 15 single nucleotide polymorphisms (SNPs) significantly associated with days to heading identified by genome-wide association study (GWAS) (-log₁₀(*P*) > 5 and MAF > 5%).** Significant SNPs identified in Group A (Blue) and Group B (Red) are indicated, with selected SNPs marked by a highlighter.

**Table 2. Genetic information of single nucleotide polymorphisms (SNPs) identified through genome-wide association study (GWAS).**

| Accessions | SNP | Chr | Position | Model and *p*-value | | PVE (%) | Allele |
|---|---|---|---|---|---|---|---|
| **Group A** | **AX-95222044** | 4B | 598261950 | BLINK | 3.91E-19 | 1.91 | G/A |
| | | | | SUPER | 3.91E-19 | 1.91 | |
| | | | | FarmCPU | 4.91E-19 | 2.03 | |
| **Group A** | **AX-94685526** | 2B | 167798344 | BLINK | 2.12E-06 | 0.42 | C/T |
| | | | | SUPER | 2.12E-06 | 0.42 | |
| **Group B** | **AX-94550996** | 6D | 451158577 | BLINK | 6.03E-09 | 2.08 | G/C |
| | | | | SUPER | 6.03E-09 | 2.08 | |
| **Group B** | **AX-94970315** | 2B | 26581499 | FarmCPU | 1.70E-10 | 2.34 | G/A |

PVE: proportion of variance explained

4.5 days earlier than those carrying an allele for late heading at the two SNPs (A SNP and T SNP), respectively (Fig 4). In Group B, AX-94550996 on chromosome 6D and AX-94970315 on chromosome 2B were selected. Accessions carrying the allele for early heading at AX-94550996 (C SNP) and AX-94970315 (A SNP) headed 5.4 days and 3.9 days earlier, respectively, than those with the allele for late heading (G SNP) at both loci (Fig 4).

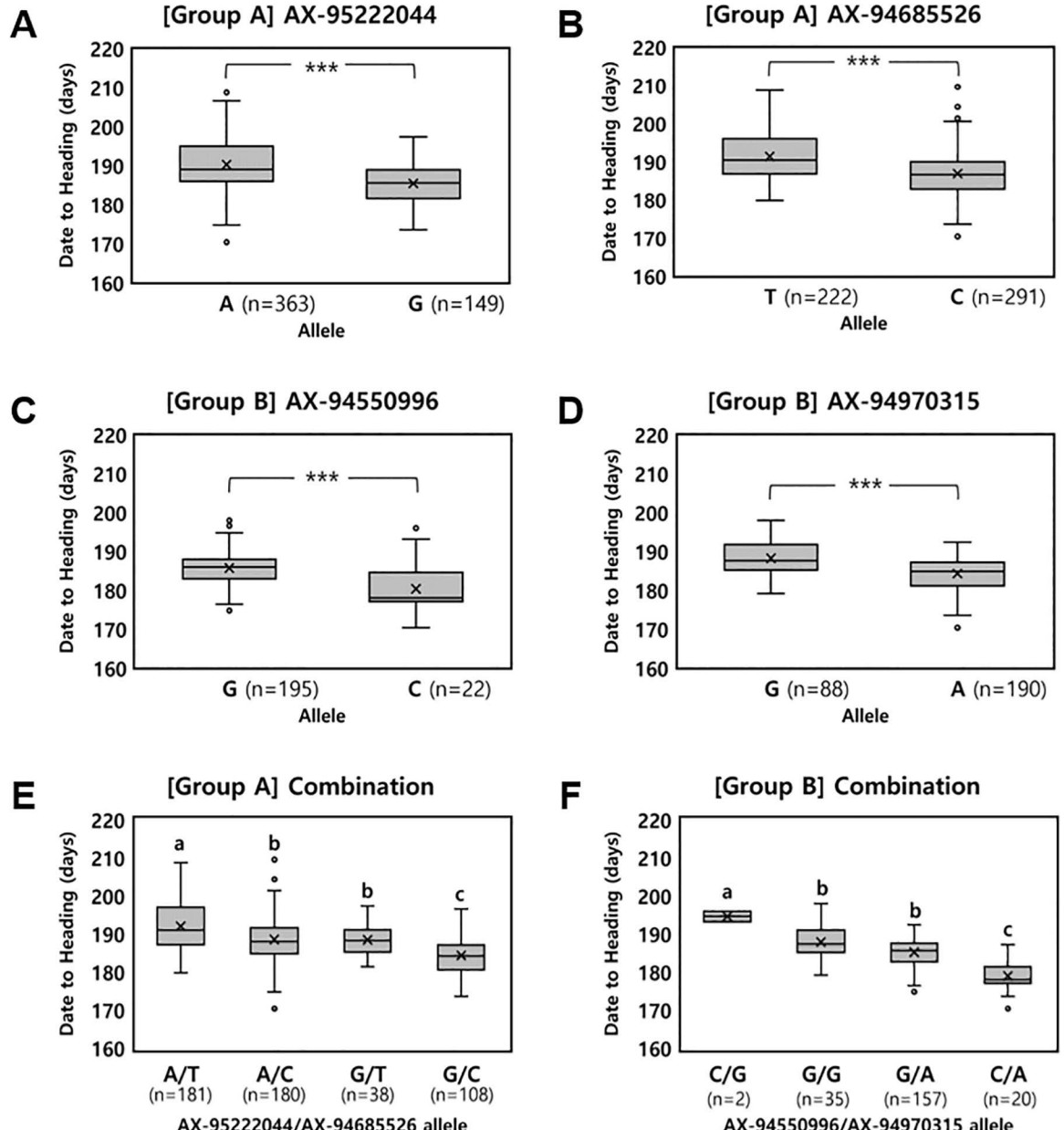

**Fig 4. Distribution of days to heading based on allelic variations of the selected single nucleotide polymorphisms (SNPs) through group analysis of wheat core collections.** In Group A, SNPs AX-95222044 (A) and AX-94685526 (B) were analyzed, while in Group B, SNPs AX-94550996 (C) and AX-94970315 (D) were analyzed. Combinations of SNPs within each group were also analyzed (E, F). *** significant at $P < 0.001$. Different letters above the boxplots indicate significant differences at $P < 0.05$ level according to Duncan's Multiple Range Test.

The heading dates of accessions possessing different allele combinations for the selected SNPs were analyzed for each group (Fig 4E, 4F). In Group A, accessions carrying the alleles for early heading at both selected SNPs had the earliest DTH of 184.1 days, while accessions carrying the alleles for late heading at both SNPs had the latest DTH of 191.8 days. The DTH of accessions carrying one allele for early heading at either of the two SNPs were both 188.3 days, indicating that pyramiding the alleles for early heading at the two loci could be effective for accelerating DTH. In Group

B, accessions carrying the alleles for early heading at both selected SNPs had the earliest DTH of 178.8 days, followed by the accessions carrying one allele for early heading at AX-94970315 (185.0 days) and the accessions carrying the alleles for late heading at both SNPs (187.7 days). However, two accessions carrying one allele for early heading at AX-94550996 (C SNP) exhibited the latest DTH of 194.4 days, indicating the possibility of genetic interaction among the two loci or effects of other heading date genes.

## Distribution and effects of the selected SNPs from GWAS

Among the top five countries of origin of the wheat core collection, Korean accessions (n=120) exhibited the highest frequency of SNPs for early heading at AX-95222044 (G SNP), AX-94685526 (C SNP), and AX-94550996 (C SNP) (Fig 5). Compared to the Korean accessions (n=120; landraces) in the core collection, Korean wheat varieties (n=40) showed even higher frequencies of alleles for early heading at AX-95222044 (G SNP), AX-94550996 (C SNP), and AX-94970315 (A SNP).

Effects of the 4 SNPs discovered through GWAS were examined in 40 Korean wheat varieties, as presented in S7 Table. For AX-94550996, 40.5% carried the allele for early heading (C SNP), exhibiting a DTH of 177.3 days, which was significantly 1.7 days earlier than those carrying the allele for late heading (G SNP). For AX-94970315, 91.9% carried the allele for early heading (A SNP), exhibiting a DTH (178.0 days) that is significantly 4.3 days earlier than those carrying the allele for late heading (G SNP). For AX-94685526, 86.5% carried the allele for early heading (C SNP), exhibiting a DTH (178.6 days) that is significantly 2.0 days later than those carrying the allele for late heading (T SNP), showing an opposite effect compared to the core collection. For AX-95222044, 67.6% carried the allele for early heading (G SNP), exhibiting a DTH (178.5 days) that is 0.6 days later than those carrying the allele for late heading (A SNP), with no significant difference in DTH.

As with the core collection, considering the four SNP genotypes selected through GWAS, the Korean varieties were divided into eight combinations (S8 Table). One combination that was not present in the core collection was identified, and the remaining seven combinations were among the top eight combinations with the earliest heading dates in the core collection. Differences in heading dates occurred between the Korean varieties and core collections for common genotypic combinations, likely owing to the small number of accessions in each genotypic combination and their different genetic backgrounds.

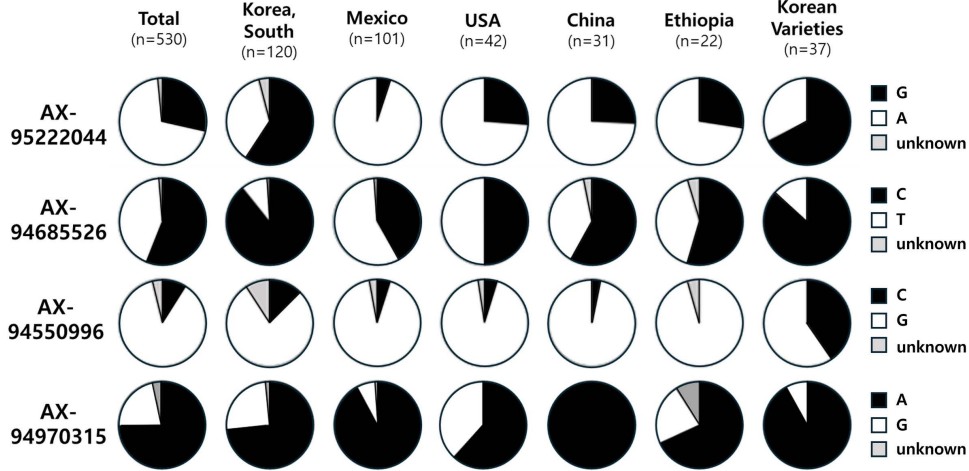

**Fig 5. Allele distribution of selected single nucleotide polymorphisms (SNPs) in the wheat core collections (n = 530) by geographic origin and Korean wheat varieties (n = 37).** Black and white indicate alleles for early heading and late heading, respectively.

## Discussion

The present study explored the genetic loci involved in variations in the DTH of wheat under autumn-sowing conditions in Korea. Using a wheat core collection, we analyzed the genotypes of previously reported major heading date genes and conducted GWAS. Heritability of the heading date phenotype in the core collection over five years was high at 0.89, which was consistent with previous findings [61,62]. This result suggests that the use of molecular markers to select early-heading wheat varieties would be efficient. Among the top five countries comprising the wheat core collection investigated herein, Korean accessions composed of landraces had the earliest heading dates, and Korean varieties headed even earlier. This result suggests that wheat breeding in Korea is directed toward developing early heading cultivars to facilitate the prevalent rice-wheat double cropping system in Korea and that the alleles for early heading might have accumulated in major heading date genes [63].

In the core collection, accessions carrying winter-type alleles for *VRN-A1* and *VRN-B1* exhibited earlier heading than those with spring-type alleles. Furthermore, globally recognized early-heading Korean landraces and varieties predominantly carry winter-type alleles for these genes. In this study, wheat was cultivated under autumn-sowing and overwintering conditions. Therefore, it is inferred that the winter-type alleles promoted early heading owing to the vernalization effect from low temperatures, whereas the spring-type alleles delayed heading, potentially due to unnecessary exposure to extended cold conditions [64]. Among the *PPD-1* genes, only the genotypes of *PPD-D1* exhibit a significant effect on DTH in the wheat core collections. This was consistent with the results of Pugsely [19] indicating that *PPD-D1* was the most influential among photoperiod-related *PPD-1* genes. In our study, *PPD-D1* exerted the most significant influence on DTH. This result is consistent with the findings of Grogan et al. [27], who indicated that the heading date in winter wheat is more strongly influenced by photoperiod loci than by vernalization loci. Notably, the effect of *VRN-D1* was also significant, albeit less pronounced than that of *PPD-D1*. Overall, these results suggest that under autumn-sowing conditions, where vernalization requirements are typically satisfied, photoperiod genes play a more dominant role in determining heading time [1,25,26]. In our study, all Korean varieties carried *vrn-A1, Ppd-A1b*, and *Ppd-D1a* alleles, aligning with previous studies that indicated limited variability in the genetic composition of the heading date-related genes *VRN-1* and *PPD-1* in Korean wheat varieties and accessions and that the previously reported heading date genes have limited ability to explain DTH variations among Korean wheat varieties [63,65–67].

The GWAS conducted on DTH utilized a wheat core collection comprised of genetically diverse accessions suitable for analysis [49,68]. The GWAS of 518 accessions (Group A) and 231 accessions (Group B, possessing the same alleles as the Korean varieties) identified nine and six non-overlapping SNPs significantly associated with DTH. Considering the practical impact on DTH, AX-95222044 and AX-94685526 were selected in Group A, and AX-94550996 and AX-94970315 were selected in Group B. The markers AX-95222044 and AX-94970315 are associated with the heading date in wheat. Specifically, in a study utilizing the wheat core collection, as in our study, GWAS was conducted on various agricultural traits, including heading date [69]. The results revealed that AX-95222044 exhibits a significant association with heading date ($-\log_{10} (P)$ = 3.09). Additionally, in another study involving a population derived from crosses between 54 Synthetic Hexaploid Wheat (SHW) lines and nested association mapping (NAM) parents, QTL analysis was performed on the heading date phenotype using 35K SNP chip data, identifying QFt.niab-2B.1 with the peak marker AX-94970315 ($-\log_{10}(P)$ = 6.34) [70]. In the present study, the four SNPs identified through GWAS exhibited individually minor but significant effects on DTH (Table 2). These minor SNPs are believed to play an important role in regulating the complex quantitative trait of heading date in wheat [71]. Therefore, incorporating these loci into breeding programs can complement the effects of *VRN* and *PPD* genes, which can enable the adjustment of heading dates to specific environmental and cultivation conditions, ultimately improving wheat adaptability and productivity. However, contrary to our expectations, *PPD-D1* was not identified in GWAS using Group A, despite the *PPD-D1* genotype exhibiting the most significant effects on DTH and the inclusion of AX-94609980 (2D: 33,953,727) within *PPD-D1* (*TraesCS2D02G079600*; 2D: 33,952,488–33,955,629) in the GWAS analysis. This result may be due to the skewed

genotype distribution of SNP AX-94609980, of which the T SNP was predominant (94.8%) in the GWAS panel, reducing statistical power.

In the present study, analysis of the genotype distribution of SNPs selected based on the country of origin in the wheat core collection revealed that among the top five countries comprising the wheat core collection, the Korean accession, which exhibited the earliest heading, demonstrated a higher proportion of genotypes for early heading than accessions from other countries. Moreover, the proportion of genotypes with an earlier heading date tended to increase in Korean varieties compared to that in Korean accessions. This result suggests that the breeding of Korean varieties has advanced the heading date through the accumulation of genetic types with earlier heading of selected SNPs. Moreover, analysis of the selected SNP genotypes in Korean varieties revealed that the two SNPs selected through GWAS in Group B, which reflected the genotypes of Korean varieties, were more suitable for explaining the variation in heading dates of the Korean varieties. Therefore, breeding efficiency could be enhanced by utilizing the four selected SNPs along with the existing heading date genes as selection markers under autumn-sowing conditions in Korea.

All four SNPs were located in the coding sequence (CDS) of specific genes. AX-95222044, located in the CDS of *TraesCS4B02G307900*, may potentially induce a loss-of-function owing to the creation of an early stop codon by the A allele. *TraesCS4B02G307900* exhibits high expression in parts of the spike and during early spike development, booting, and flowering stages, according to the Wheat Expression Browser prediction (https://www.wheat-expression.com). AX-94685526, located in the CDS of *TraesCS2B02G192300*, induces amino acid changes. However, no specific expression related to heading was observed. AX-94550996, located in the CDS of *TraesCS6D02G358000*, induces amino acid changes depending on the genotype. This gene exhibits high expression in the central axis of leaves and flowers, and its *Arabidopsis* ortholog *AT5G54390* influences flowering time [72]. AX-94970315, located in the CDS of *TraesCS2B02G054700*, does not cause amino acid changes based on the genotype. Therefore, further research using additional genetic analysis populations would be required to validate the effects of these four heading date-associated loci and to identify the causative genes.

## Supporting information

**S1 Fig. Mean temperature and precipitation over the test years (2018–2022).** ─●─ and ▦ indicate the mean temperature and precipitation for the test years, and ─○─ and ▢ indicate the mean temperature and precipitation for a long-term average (1991–2020). I, II, III, IV, V, VI, and VII represent the emergence, regeneration, tillering, elongation, heading, milky, and maturation stages, respectively.
(TIF)

**S2 Fig. Correlation between test year days to heading (2018–2022) and their average.** (A) Wheat core collections (n = 530). (B) Korean wheat varieties (n = 40). *** significant at $P < 0.001$.
(TIF)

**S3 Fig. Principal component analysis results and the distribution of accessions of wheat core collections.** (A) x: PC1, y: PC2, (B) x: PC1, y: PC3.
(TIF)

**S4 Fig. Manhattan plots and QQ plots for different genome-wide association study (GWAS) models.** The results correspond to the application of MLM, CMLM, and MLMM to Group A.
(TIF)

**S1 Table. Genetic accessions used in this study according to the country of origin.**
(DOCX)

**S2 Table. List of the PCR primers used for the analysis of allelic variations in *VRN-1* and *PPD-1*.**
(DOCX)

**S3 Table. Heritability estimation of days to heading in wheat core collections.**
(DOCX)

**S4 Table. Grouping of wheat core collections based on the *VRN-1* and *PPD-1* genotypes.**
(DOCX)

**S5 Table. Distribution and days to heading (DTH) based on the *VRN-1* and *PPD-1* genotypes in the wheat core collections by origin and Korean wheat varieties.**
(DOCX)

**S6 Table. Single nucleotide polymorphisms (SNPs; -$\log_{10}(P)$ > 5 and MAF > 5%) identified through genome-wide association study (GWAS).**
(DOCX)

**S7 Table. Distribution and days to heading (DTH) based on selected single nucleotide polymorphism (SNP) in the wheat core collections by origin and Korean wheat varieties.**
(DOCX)

**S8 Table. Grouping of Korean wheat varieties based on the selected single nucleotide polymorphism (SNP) genotypes.**
(DOCX)

## Acknowledgments

Not applicable.

## Author contributions

**Conceptualization:** Yurim Kim, Chul Soo Park, Ki-Chang Jang, Changhyun Choi.

**Data curation:** Yurim Kim, Jun Yong Choi, Changhyun Choi.

**Formal analysis:** Yurim Kim, Myoung-Goo Choi, Youngjun Mo.

**Investigation:** Yurim Kim.

**Methodology:** Yurim Kim.

**Project administration:** Yurim Kim.

**Resources:** Suk-Jin Kim, Chon-Sik Kang, Changhyun Choi.

**Supervision:** Youngjun Mo.

**Visualization:** Yurim Kim.

**Writing – original draft:** Yurim Kim.

**Writing – review & editing:** Yurim Kim, Myoung-Goo Choi, Myoung Hui Lee, Chuloh Cho, Youngjun Mo.

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
