## [Decision Letter · Decision Letter 0]

18 Nov 2024

PONE-D-24-37356Genome-wide association study to identify the genomic loci associated with wheat heading date variation under autumn-sowing conditionsPLOS ONE

Dear Dr. Choi,

Thank you for submitting your manuscript to PLOS ONE. After careful consideration, we feel that it has merit but does not fully meet PLOS ONE’s publication criteria as it currently stands. Therefore, we invite you to submit a revised version of the manuscript that addresses the points raised during the review process.

There are several comments related to methodology and results that need to be addressed. Please go through the comments for further details

We look forward to receiving your revised manuscript.

Kind regards,

Harsh Raman, Ph.D

Academic Editor

PLOS ONE

Journal Requirements:

2. We note that your Data Availability Statement is currently as follows: “All relevant data are within the manuscript and in Supporting Information files.”

Please confirm at this time whether or not your submission contains all raw data required to replicate the results of your study. Authors must share the “minimal data set” for their submission. PLOS defines the minimal data set to consist of the data required to replicate all study findings reported in the article, as well as related metadata and methods (https://journals.plos.org/plosone/s/data-availability#loc-minimal-data-set-definition). For example, authors should submit the following data: - The values behind the means, standard deviations and other measures reported; - The values used to build graphs; - The points extracted from images for analysis. Authors do not need to submit their entire data set if only a portion of the data was used in the reported study. If your submission does not contain these data, please either upload them as Supporting Information files or deposit them to a stable, public repository and provide us with the relevant URLs, DOIs, or accession numbers. For a list of recommended repositories, please see https://journals.plos.org/plosone/s/recommended-repositories. If there are ethical or legal restrictions on sharing a de-identified data set, please explain them in detail (e.g., data contain potentially sensitive information, data are owned by a third-party organization, etc.) and who has imposed them (e.g., an ethics committee). Please also provide contact information for a data access committee, ethics committee, or other institutional body to which data requests may be sent. If data are owned by a third party, please indicate how others may request data access.

Additional Editor Comments:

There are several issues which needs to be addressed. I have attached my comments

Reviewers' comments:

Reviewer's Responses to Questions

**Comments to the Author**

1. Is the manuscript technically sound, and do the data support the conclusions?

Reviewer #1: Yes

2. Has the statistical analysis been performed appropriately and rigorously? 

Reviewer #1: Yes

3. Have the authors made all data underlying the findings in their manuscript fully available?

Reviewer #1: Yes

4. Is the manuscript presented in an intelligible fashion and written in standard English?

Reviewer #1: Yes

5. Review Comments to the Author

Reviewer #1: The paper is well written. The introduction sets out the reasons for the study, but the methods and data do not adequately answer the questions being posed. A high percentage of alleles at known genetic loci for heading date in wheat are described as unknown. These should be further investigated, as they could represent novel alleles or genes, and would add quality to the manuscript. I suggest to do this and resubmit.

6. PLOS authors have the option to publish the peer review history of their article (what does this mean? ). If published, this will include your full peer review and any attached files.

**Do you want your identity to be public for this peer review?** For information about this choice, including consent withdrawal, please see our Privacy Policy .

Reviewer #1: **Yes: ** Livinus Emebiri

---

## [Author Response · Author response to Decision Letter 1]

3 Jan 2025

Response to Reviewer’s comments

We sincerely appreciate the reviewer’s valuable comments, which have greatly contributed to improving the quality of our manuscript. Below, we provide our response to the reviewer’s comments.

Major revisions

1. In general, the manuscript would be of limited interest, as the findings are supportive of well known information. The study identified a percentage of alleles at Vrn and Ppd loci that were listed as ‘unknown’ and with no record of the associated DTH. These could represent rare alleles and should be investigated further to deliver quality science, otherwise the content of paper is only of regional interest.

- As the reviewer pointed out, our study deals with well-known loci, which may limit its novelty. However, our study employed a differentiated strategy to identify new genetic markers influencing heading date beyond the well-known loci using GWAS. Specifically, we selected resources from a wheat core set that had the same VRN1 and PPD1 genotypes as those of Korean wheat varieties, which are known globally for their early heading characteristics. Using this approach, we were able to discover new SNP markers associated with heading date variation. Additionally, we revised the Abstract and Discussion sections of the manuscript to better highlight this differentiated methodology.

- I believe that this paper not only proposes an innovative experimental approach in GWAS analysis but also provides valuable information for the development of more effective heading date markers applicable to countries such as Japan and China.

- Despite repeated PCR experiments performed multiple times to confirm the Vrn and Ppd genotypes, resources with unresolved genotypes are still classified as 'unknown.' I believe this issue cannot be resolved using the same approach. To address this problem, it will be necessary to sequence the full nucleotide sequences of the Vrn and Ppd loci and design various primer combinations based on this information. This could present a new and independent challenge.

2. The paper is well written. The introduction sets out the reasons for the study, but the methods and data do not adequately answer the questions being posed. A high percentage of alleles at known genetic loci for heading date in wheat are described as unknown. These should be further investigated, as they could represent novel alleles or genes, and would add quality to the manuscript. I suggest to do this and resubmit.

- I agree with the reviewer's opinion. Therefore, we conducted a statistical analysis of the number of days to heading by allele for the newly discovered genetic marker loci, as well as for each of the already known heading genes (Vrn and Ppd), as presented in Table 1 and Table 2, respectively, in line with the reviewer’s suggestion. Additionally, we incorporated explanations into the Results and Discussion sections, highlighted in blue text.

Minor revisions

2 page 32 line : Pl present data

Response: Corrected

2 page 36 line : Which loci-describe here

Response: Specifically provided information about the loci in the abstract.

3 page 56 line : Describe this

Response: Corrected

5 page 99 line : 40 are part of 530?

Response: Indicated that the 530 wheat core collection resources and the 40 Korean varieties are separate.

5 page 101 line : This does not equates to 362

Response: The numbers provided in the manuscript represent the major contributors to the 362 accessions collected from 47 countries. Smaller contributions from other countries were not explicitly listed to maintain conciseness, but they are included in the total count of 362 accessions.

5 page 110 line : Not self explanatory

Response: Corrected

6 page 113 line : What this means, regeneration stage?

Response: Added explanations for the terms.

7 page 138 line : Have authors used VRN-A1, VRN-B1, and VRN-D1 data for GWAS?

Response: The VRN-1 data was indirectly utilized in GWAS, as the second GWAS (Group B) was performed using accessions that possess the vrn-A1 allele, reflecting the allelic composition of Korean wheat varieties.

7 page 145 line : What were the values for heterozygosity? + and - 3 SD?

Response: Provided the values.

11 page table : Average?, Author should have conducted MET analysis

Response: Reanalyzed the data to present stability index by genotype and revised the relevant content in the manuscript accordingly.

page 237 line : What this means, in relation to population structure?

Response: The percentages of the principal components (PC1, PC2, and PC3) indicate the proportion of total genetic variation explained by each component. PC1, explaining 6.0% of the variance, captures the primary axis of genetic variation, while PC2 and PC3, explaining 3.8% and 2.7% respectively capture additional axes of variation. These values reflect the genetic structure within the population.

14 page 273 line : ?

Response: Corrected

15 page table : Provide R2 values-phenotypic or genetic variance accounted

Response: Added proportion of variance explained (PVE).

30 page 402 line : data not shown

Response: Corrected

In addition, some revisions were made to improve the clarity of the manuscript.

---

## [Editor Report · Decision Letter 1]

19 Feb 2025

PONE-D-24-37356R1Genome-wide association study to identify the genomic loci associated with wheat heading date variation under autumn-sowing conditionsPLOS ONE

Dear Dr.  Choi,

Thank you for submitting your manuscript to PLOS ONE. After careful consideration, we feel that it has merit but does not fully meet PLOS ONE’s publication criteria as it currently stands. Therefore, we invite you to submit a revised version of the manuscript that addresses the points raised during the review process.

Lines 257: -… located on the short arm of chromosome 2A, in close proximity to PPD-A1 (TraesCS2A01G081900; 2A: 36,936,362–36,938,400). The search for 'TraesCS2A01G081900' returned no results. Please recheck and submit these manuscriptPlease ensure that your decision is justified on PLOS ONE’s publication criteria  and not, for example, on novelty or perceived impact.

We look forward to receiving your revised manuscript.

Kind regards,

Harsh Raman, Ph.D

Academic Editor

PLOS ONE

Journal Requirements:

Additional Editor Comments:

I have gone thru the manuscript and the authors addressed queries raised by the reviewer. I am happy to accept this manuscript if author address the following comment

Lines 257: -… located on the short arm of chromosome 2A, in close proximity to PPD-A1 (TraesCS2A01G081900; 2A: 36,936,362–36,938,400).

The search for 'TraesCS2A01G081900' returned no results. Please, check and correct.

---

## [Author Response · Author response to Decision Letter 2]

13 Mar 2025

Response to Reviewer’s comments

We sincerely appreciate the reviewer’s additional comments, which have helped us further improve our manuscript. Below, we provide our response to the reviewer’s comments.

Minor revisions

1. Lines 257: -… located on the short arm of chromosome 2A, in close proximity to PPD-A1 (TraesCS2A01G081900; 2A: 36,936,362–36,938,400). The search for 'TraesCS2A01G081900' returned no results. Please recheck and submit these manuscript.

Response: The reason why the gene TraesCS2A01G081900 (2A: 36,936,362–36,938,400) could not be found is that it was annotated in the IWGSC 1.0 assembly version. This information can be verified in Reference 9 of our manuscript. In the IWGSC 2.1 assembly version, the gene has been renamed to TraesCS2A02G081900. Therefore, we have updated our manuscript accordingly, replacing TraesCS2A01G081900 with TraesCS2A02G081900 and specifying the corresponding genomic location

---

## [Editor Report · Decision Letter 2]

19 Mar 2025

Genome-wide association study to identify the genomic loci associated with wheat heading date variation under autumn-sowing conditions

PONE-D-24-37356R2

Dear Dr. Choi,

We’re pleased to inform you that your manuscript has been judged scientifically suitable for publication and will be formally accepted for publication once it meets all outstanding technical requirements.

Kind regards,

Harsh Raman, Ph.D

Academic Editor

PLOS ONE
---

## [Editor Report · Acceptance letter]

PONE-D-24-37356R2

PLOS ONE

Dear Dr. Choi,

I'm pleased to inform you that your manuscript has been deemed suitable for publication in PLOS ONE. Congratulations! Your manuscript is now being handed over to our production team.

Kind regards,

on behalf of

Dr. Harsh Raman

Academic Editor

PLOS ONE